# Spectral Optimization of White LED Based on Mesopic Luminance and Color Gamut Volume for Dim Lighting Conditions

**Hung-Chung Li** [1,*] , **Pei-Li Sun** [2], **Yennun Huang** [1] and **Ming Ronnier Luo** [3]

[1]  Research Center for Information Technology Innovation, Academia Sinica, Taipei 11529, Taiwan; yennunhuang@gmail.com

[2]  Graduate Institute of Color and Illumination Technology, National Taiwan University of Science and Technology, Taipei 10607, Taiwan; plsun@mail.ntust.edu.tw

[3]  State Key Laboratory of Modern Optical Instrumentation, Zhejiang University, Hangzhou 310058, China; m.r.luo@zju.edu.cn

*  Correspondence: pony780210@citi.sinica.edu.tw

**Abstract:** The study aims to propose an approach of white LED spectral optimization based on mesopic luminance and color gamut volume for dim lighting conditions. Three optimal white LED spectra with relatively higher mesopic luminance and color gamut volume, the highest mesopic luminance, and the largest gamut volume are recommended for reducing energy consumption and enhancing color perception and recognition of human eyes. The theoretical simulation shows that the spectra with higher correlated color temperatures (CCT) and S/P-ratio increase the mesopic luminance and also extend the range of color gamut with the decreasing of lighting level. An evaluation model is developed to faster predict mesopic luminance, color gamut volume, and S/P ratio for lighting applications.

**Keywords:** white light-emitting diodes; mesopic vision; spectral optimization; color gamut estimation

## 1. Introduction

The luminance level of night-time conditions normally falls into a mesopic visual range within 0.001–3 cd/m$^2$. From previous studies for mesopic range, two methods, which were brightness matching and visual performance-based approaches, were mainly used for establishing the mesopic sensitivity [1]. The mesopic spectral luminous efficiency could be described in terms of a linear combination of the photopic $V(\lambda)$ and scotopic $V'(\lambda)$ spectral luminous efficiency functions. The system was formed as Equations (1) and (2), where $M(m)$ was a normalizing function to make the maximum value of $V_{mes}(\lambda)$ equal to 1, $V_{mes}(\lambda_0)$ was the value of $V_{mes}(\lambda)$ at 555 nm, $L_{mes}$ represented the mesopic luminance, and $L_e(\lambda)$ was the spectral radiance ($W \cdot m^{-2} \cdot sr^{-1} \cdot nm^{-1}$). Depending on the upper and lower luminance limits, m was set to 0 and 1, which corresponded to the scotopic and photopic spectral luminous efficiency.

$$M(m)V_{mes}(\lambda) = mV(\lambda) + (1-m)V'(\lambda), \; for \; 0 \le m \le 1 \tag{1}$$

$$L_{mes} = \frac{683}{V_{mes}(\lambda_0)} \int V_{mes}(\lambda) L_e(\lambda) d\lambda \tag{2}$$

In previous studies on mesopic vision, models of mesopic luminous efficiency, such as X-model [2] and MOVE-model [3], were mainly derived for calculating mesopic luminance. The X-model was derived from a reaction time experiment in the mesopic range. In the X-model, a parameter X was

used to characterize the ratio between photopic and scotopic luminous efficacy at any luminance. The MOVE-model was proposed by European research consortium MOVE (Mesopic Optimization of Visual Efficiency) as a recommended model for mesopic luminance of road lighting applications. The model was derived from a series of experiments related to visual task performance such as achromatic contrast threshold, reaction time, and recognition threshold. In spite of the similar forms of the X-model and MOVE-model, the major differences of the two models are the upper luminance limits of the mesopic range, which is considered too low for the X-model and too high for the MOVE-model. Hence, a modified MOVE-model was later proposed with an appropriate upper luminance limit [4]. A CIE (International Commission on Illumination) recommended mesopic photometric system, CIE TC 1-58 model, also has been successfully used in mesopic photometry for evaluating LED outdoor lighting. Although the models could adequately predict the luminance for monochromatic lights, the effects of chromatic lights on visual performance were not yet taken into account. By applying mesopic luminous efficiency functions, there are significant differences in the luminous efficiency scale between LED light sources with different correlated color temperatures [5]. The visual performance for night-time driving conditions was investigated by carrying out three visual tasks, achromatic threshold, reaction time, and perception of details included when driving a car. With decreasing luminance level from 1 to 0.01 cd/m$^2$, the visual performance of achromatic threshold and reaction time dropped obviously, and also presented the Purkinje shift. On the basis of their measurements to visual performance, the spectral sensitivity function could be well described with a mesopic model [6]. The X-model was compared to a color model, including inputs from both the L-M color-opponent mechanism and the S-cones to characterize spectral sensitivity with reaction time. The study showed that the second model mentioned above was a better fit for the visual data, whereas the X-model did not account for the reaction time of a stimulus [7]. Viikari suggested a new modified MOVE-model with a suitable upper luminance limit, which was 5 cd/m$^2$ to meet the street and road lighting conditions in reality for different weather conditions. The paper tested the X-model, MOVE-model, and modified MOVE-model by use of visual data acquired from European universities. As a result, the modified MOVE-model described the data best with over half of the situations [8]. Jin found that both background luminance and spectra power distribution will affect mesopic luminous efficiency. In other words, the optimal luminous efficiency can be achieved by varying the spectrum distribution of LED sources under a specific background [9]. Li recommended the coefficient *B* derived for efficiently calculating the mesopic equivalent illumination ($E_{mes}$) from photopic illumination ($E_p$) for the light sources with different correlated color temperatures (CCT) under mesopic conditions as shown in Equation (3). The findings suggested that the mesopic illumination modified by coefficient *B* was close to meeting the human visual perception. Under the same luminance condition, the coefficient *B* increased with the light source possessing higher CCT, which was beneficial to road lighting [10].

$$E_{mes} = B \cdot E_p \tag{3}$$

Two methods inclusive of the Adaptation and Source SPD (spectral power distribution) were recently proposed for simply measuring the mesopic luminance and tested with real road surface spectral reflectance in Uchida's study. The main difference between these two methods was that the Source SPD method considered the S/P-ratio (scotopic/photopic ratio) for both test points and the adaptation field with light source's S/P-ratio. In contrast, the Adaptation SPD method only approximately estimated the S/P-ratio of test points with the S/P-ratio of the adaptation field. The simulation showed that the Source SPD method caused a significant error of road surface spectral reflectance but could reduce inaccuracy with a correction. In addition, the Source SPD method was not adequate for lighting applications in a scene with multiple light source types [11]. A simulation also conducted by Uchida involved four factors, which were the area of measurement (AOM), surrounding luminance effect (SLE), luminance distribution (LD), and eye movement (EMs) of observers to determine adaption luminance for mesopic photometry. From the comparison between the simulation method and two predictors for complex luminance distributions (LD), it pointed out

that the adaption luminance could be well predicted with the average luminance of the AOM method except for the condition that has high luminance sources around the AOM [12].

In general, a light source with higher correlated color temperature is provided with a higher S/P-ratio [13]. If a light source has a higher S/P-ratio, visual brightness would be relatively higher in the scotopic conditions, which is likely to enhance night-time visual performance and beneficial to mesopic design [14]. However, if a light source has stronger spectral radiance in about 530 nm, it should be relatively brighter in mesopic vision, but S/P-ratio cannot depict it. Wu discovered a solution for improving the parameters related to the mesopic application, including S/P-ratio and the mesopic luminance of a light source. In their experiment, an optical power ratio algorithm was adopted to select appropriate spectral power distribution (SPD) for their optimal purpose [15].

The CIE proposed Color Rendering Index (CRI) metric compares color differences of eight Munsell color samples illuminated by the light source and a reference illuminant (daylight for CCT ≥ 5000 K; blackbody for CCT < 5000 K) with the same CCT. Due to the visual judgment affected by high chroma colors and the chroma of the color samples used by the CRI are not high enough, previous studies showed that the CRI score does not have an excellent correlation to the visual preference of saturating LED light sources. To solve this problem, the color quality scale (CQS), which uses high chroma test samples and non-line color difference scaling, was proposed to heighten the correlation. Zan developed a spectral optimization model to acquire the most appropriate SPD for unique mesopic luminous efficacy of white LED consisted of a blue-chip, yellow, and red quantum dots with the ideal color rendering properties. From their discovery, the optimal full width at half maximum (FWHM) of each color was 30 nm, which could achieve the maximum value of mesopic luminous efficiency. The optimized peak wavelength and reasonable colorimetric performances of each color component with CRI ≥ 70 and CQS ≥ 60 as well as CRI ≥ 85 and CQS ≥ 85 for maximizing the mesopic luminous efficiency at CCTs from 2700 K to 45,000 K were here recommended [13]. The CRI refers to the general Color Rendering Index, which is a metric to judge the color performance of a light source. Lei designed desirable trichromatic and tetrachromatic SPDs of white LED light sources that have high luminous efficacy and acceptable color rendering. An optimal spectrum for trichromatic white LED with a combination of wavelengths at 460, 540, and 615 nm provided the best luminous efficacy, 336 lm/W [16]. Jiandong discussed the visual performance by implementing a dynamic experiment in reaction time with high, middle, and low color temperatures, and the results implied that the reaction time decreased with the raising of background brightness and contrast and the high color temperature LED [17]. Gao modified the parameters *a* and *b* to avoid no solution or multi-solutions and proposed the Bisection-Newton method for faster convergence in the CIE system of mesopic photometry MES2. They also provided an initial guess strategy later and recommended that the S/P ratio should be smaller than a constant value of about 18.1834, which is better for convergence when the value of the S/P ratio is too large [18,19]. An alternative approach of iteration method was established based on a Rational Taylor function for modeling with a small average error of about 0.28% compared with the results utilizing iteration [20]. From the study of Li, a white LED consisting of four-hump quantum dot light-emitting diodes could improve the luminous and the visual performance with the S/P ratio of 2.52, the luminous efficacy of 460.78 lm/W, the CCT of 5454 K, and the CRI of 90.3 [21]. Zheng adopted the Monte Carlo and genetic algorithm to find an optimal white LED spectrum on the basis of the S/P ratio and CRI for the mesopic vision, where the S/P ratio, CCT, and CRI were 2.064, 4981 K, and 93.9, respectively [22]. Xiao optimized the white LED spectra with the combinations of blue, green, and orange components to reach the ideal luminous efficacy and the $R_f$ values of IES (Illuminating Engineering Society) color fidelity over 70, 80, and 90 simultaneously [23]. A functional model based on the chromaticity-guided was built for lighting applications of the trichromatic light source in both photopic and mesopic conditions [24]. Maksimainen developed a measurement technology by use of a calibrated digital camera to acquire pixel-wise S/P ratio and further calculated the mesopic luminance [25]. Shin proposed a mesopic color appearance model, which could be used to simulate

color appearance in the mesopic range as well as photopic and scotopic conditions by adding rod intrusion to the two-stage model [26,27].

Most night-time outdoor and traffic lighting scenarios are in the mesopic range. However, the majority of the white LED spectrum is optimized based on luminous efficiency and color rendering index mostly in the photopic vision that shows a wider color gamut than in mesopic vision. As the color gamut in the mesopic conditions are relatively small, the general color rendering index is not applicable to evaluate color fidelity. Therefore, an optimization of the white LED spectrum based on color gamut is essential to raise the ability of color recognition under constant contrast conditions for lighting application.

The present study includes two major parts: (1) optimization of white LED spectra, (2) building an evaluation model under dim lighting conditions for lighting applications. Different peak wavelength positions and density ratios are tested to obtain optimal spectra and see whether the use of a neural network can successfully predict the mesopic luminance, color gamut volume, and S/P ratio.

## 2. Methods

### 2.1. Test Spectra

Three models [16,22,23] of spectra denoted as MS1, MS2, and MS3 for the trichromatic white LED were employed and simulated by the Gaussian-like functions with normalization as Equation (4) for blue-chip, green, and red/orange phosphor of a phosphor-coated white LED (pc-WLED).

$$g(x) = \exp\left(\frac{-0.5 \cdot (x - x_0)^n}{alpha^n}\right), \, alpha = \text{FWHM} \cdot \frac{\sqrt{2\ln(2)}}{2} \tag{4}$$

where $x$, $x_0$, and *alpha* represent the wavelength from 380 to 780 nm with 5 nm interval, peak wavelength, and a function of spectral half-width for each component, respectively, and $n$ is set as 2 in the study.

A spectral database in which the Duv is smaller than 0.0054 [13,23] at CCTs of 2000–8000 K is established from the peak wavelength shifts with 5 nm spectral interval and the variation of power density ratios for each component. The spectral parameters of each model are listed in Table 1. The full width at half maximum (FWHM) of blue, green, and red/orange LEDs are 25, 36, 13 nm for MS1, 22, 73, 88 for MS2, and 25, 70, 70 for MS3, respectively. In total, 5273, 45,382, and 39,076 records of MS1, MS2, and MS3 are obtained for spectral optimization.

**Table 1.** Spectral parameters for test.

|        | Components          | FWHM (nm)    | Wavelength Shifts (nm)        | Density Ratios | Records |
| ------ | ------------------- | ------------ | ----------------------------- | -------------- | ------- |
| **MS1** | Blue, green, red    | (25, 36, 13) | B: 450–480                    | 0.1–1.0        | 5273    |
| **MS2** | Blue, green, red    | (22, 73, 88) | G: 500–550                    | 0.1–1.0        | 45,382  |
| **MS3** | Blue, green, orange | (25, 70, 70) | R: 605–645<br>Or: 580–620     | 0.1–1.0        | 39,076  |

### 2.2. Mesopic Luminance and Color Appearance

To calculate mesopic luminance, the S/P-ratio of a light source is essential. It is the scotopic lumens divided by photopic lumens as Equation (5).

$$\text{S/P} = \frac{L_s}{L_p} = \frac{\int_{380}^{780} k_s V'(\lambda) S(\lambda) d\lambda}{\int_{380}^{780} k_p V(\lambda) S(\lambda) d\lambda} \tag{5}$$

where $k_s$ and $k_p$ are 1699 and 683 lm/W, respectively, which are the peak values of luminous efficiency in photopic and scotopic ranges, and $S(\lambda)$ is the SPD of a light source.

The coefficient and the mesopic luminance $L_{mes}$ can be obtained by Equations (6)–(8), which is an iterative method. $L_p$ is the photopic luminance, $L_s$ is scotopic luminance, and $V'(\lambda_0)$ represents the scotopic luminous efficiency at 555 nm, where the value is 683/1699, $n$ is the number of iterative steps, $a$ and $b$ are parameters which are 0.7670 and 0.3334, respectively.

$$m_0 = 0.5, \tag{6}$$

$$L_{mes,n} = \frac{m_{(n-1)}L_p + (1 - m_{(n-1)})L_s V'(\lambda_0)}{m_{(n-1)} + (1 - m_{(n-1)})V'(\lambda_0)}, \tag{7}$$

$$m_n = a + b \log_{10}(L_{mes,n}), \ for \ 0 \leq m_n \leq 1 \tag{8}$$

Equation (7) can be rewritten as Equation (9), showing that S/P-ratio and the photopic luminance are the primary factors and positive correlation with $L_{mes}$ for calculating the mesopic luminance. To reach the maximum value of mesopic luminance, one aim of the study is to optimize SPD with the highest S/P-ratio in the same $L_p$.

$$L_{mes,n} = \frac{m_{(n-1)} + (1 - m_{(n-1)})(S/P)V'(\lambda_0)}{m_{(n-1)} + (1 - m_{(n-1)})V'(\lambda_0)}L_p \tag{9}$$

The mesopic appearance model describes that the opponent process converts the cone responses from *L*, *M*, and *S* to red/green, yellow/blue opponent-color, and luminance channels denoted by *L-2M*, *L+M-S*, and *L+M*, respectively. The model considered the characteristic of the nonlinear shift in spectral luminous efficiency, reduction of saturation at low illuminance levels, and the variations of hue and chroma loci under different illuminance levels from 1000 to 0.01 lux. The experimental results showed that chroma decreases continuously with the decrease of the illuminance level until 0.01 lux. The mesopic color appearance model puts a rod intrusion varied with different illuminance levels to each channel of the two-stage color-vision model. The outputs could be acquired with Equations (10) and (12).

$$A(E) = \alpha(E)K_w((L_p + M_p)/(L_p + M_p)_w) + \beta(E)K'_w(Y'/Y'_w)^\gamma \tag{10}$$

$$r/g(E) = l(E)(L_p - 2M_p) + a(E)Y' \tag{11}$$

$$b/y(E) = m(E)(L_p + M_p - S_p) + b(E)Y' \tag{12}$$

where $A(E)$, $r/g(E)$, and $b/y(E)$, respectively, represent the outputs of the luminance, red-green, and blue-yellow channels of an illuminance *E*, and $L_p$, $M_p$, and $S_p$ represent cone responses in photopic condition. $Y'$ is the scotopic luminance where the rod signals are involved. $K_w$ and $K'_w$ are the maximum responses of luminance in photopic and scotopic conditions in which the values are 100 and 78.4, respectively. $\gamma$ is the parameter of the nonlinear relationship between photopic and scotopic luminance channels attained by the least-squares fitting to the experimental data. $\alpha(E)$, $\beta(E)$, $l(E)$, $a(E)$, $m(E)$, and $b(E)$ are the different functions of weighting coefficients under illuminance *E*. The weighting coefficients under a specific luminance level could be calculated with an interpolation process and divided by a solid angle between the range of 1000 and 0.01 lux.

## 2.3. Optimization Procedure

The optimization procedure of the white LED spectrum is shown in Figure 1. The optimal white LED SPD for specific luminance level can be obtained by the following steps: first, apply an equal-energy white spectrum and input a specific luminance level to calculate the total radiance of the spectrum in 400–700 nm range. If the luminance level is below 5 cd/m$^2$, the modified MOVE-model is

adopted to calculate the power of the spectrum. Conversely, only the photopic luminosity function is used. Second, input a series of test SPDs with normalization and acquire the absolute radiant power based on the same total radiance with the equal-energy white spectrum to calculate the photopic, scotopic, and mesopic luminances. Third, Judd modified tristimulus values $X_{Judd}$, $Y_{Judd}$, $Z_{Judd}$, and CIE $V'(\lambda)$ are obtained by the integral of test spectra, spectral reflectance, Judd modified color matching functions, and the scotopic luminous function. Then the tristimulus values are converted to the spectral responses of the cones, $L$, $M$, and $S$. The scotopic luminance $Y'$ is also calculated. Next, the $L$, $M$, and $S$ responses and luminance $Y'$ are regarded as the inputs for the Shin's mesopic color appearance model.

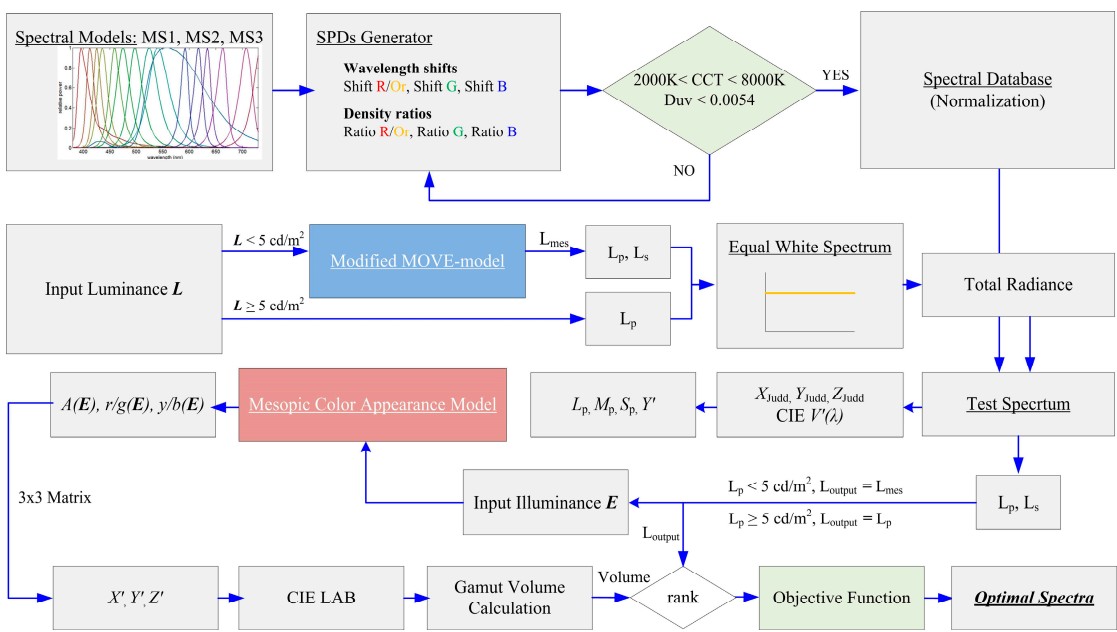

**Figure 1.** Flow chart of white LED spectrum optimization.

The mesopic/photopic luminance acquired from test SPDs shall be converted to the illuminance unit first for applying the mesopic color appearance model. After simulating the color appearance, $A(E)$, $r/g(E)$, $y/b(E)$ under a specific luminance level, the simulated tristimulus values $X'$, $Y'$, $Z'$ are obtained by a $3 \times 3$ transformation matrix and converted to CIE LAB color space for its gamut volume calculation. In the theoretical simulation, the spectral reflectances of 1269 matt Munsell color chips are used to construct a gamut boundary. The gamut volume is the summation of all tetrahedron's volume where the tetrahedra are determined by 3D Delaunay triangulation of the color samples, and the unit of gamut volume is LAB cube. Finally, according to the predicted photoic/mesopic luminance ($L_{mes}$) and the gamut volume ($GV$), the test spectra with relatively higher $L_{mes}$ and $GV$ found by achieving the maximum of the objective function as Equation (13), highest $L_{mes}$ and largest $GV$ are selected as the optimal white LED spectra under a specific luminance level.

$$Z = f(L_{mes}, GV) = \frac{L_{mes} - \min(L_{mes})}{\max(L_{mes}) - \min(L_{mes})} \cdot \frac{GV - \min(GV)}{\max(GV) - \min(GV)} \tag{13}$$

## 3. Results and Discussion

### 3.1. Optimal White LED Spectra

In the study, four luminance levels, including 10, 1, 0.3, and 0.1 cd/m$^2$ were investigated for both photopic and mesopic vision. Based on the above-mentioned white LED optimization method, the optimal white LED spectra with relatively higher $L_{mes}$ and $GV$, highest $L_{mes}$ and largest $GV$ denoted as Opt[#]1, Opt[#]2, and Opt[#]3, respectively, were derived for each luminance levels upon the same radiant

quantity condition, and the spectral information is shown in Table 2. The FWHMs of the optimal spectra are 25, 36, and 13 nm for blue, green, and red, except for Opt#2, which is 25, 70, and 70 nm in 10 cd/m$^2$.

**Table 2.** Information of optimal spectra.

| cd/m$^2$ | Peak Wavelength (nm) | | | Density Ratios | | |
|---|---|---|---|---|---|---|
| | Opt#1 | Opt#2 | Opt#3 | Opt#1 | Opt#2 | Opt#3 |
| 10 | 450, 545, 610 | 450, 550, 585 | 450, 530, 615 | 0.4, 0.5, 1.0 | 0.4, 1.0, 0.9 | 0.6, 0.5, 1.0 |
| 1 | 450, 535, 610 | 455, 550, 605 | 450, 530, 615 | 0.3, 0.4, 1.0 | 0.1, 0.2, 0.6 | 0.6, 0.5, 1.0 |
| 0.3 | 450, 530, 615 | 455, 550, 605 | 450, 510, 635 | 0.6, 0.5, 1.0 | 0.1, 0.2, 0.6 | 0.2, 0.3, 0.8 |
| 0.1 | 450, 515, 615 | 465, 510, 605 | 450, 515, 620 | 0.4, 0.5, 0.9 | 0.3, 0.5, 1.0 | 0.4, 0.5, 1.0 |

The SPDs of optimal white LED under four luminance levels are demonstrated in Figure 2. The results indicate that the peak wavelength of green phosphor shifts to the shorter wavelength with the decrease of luminous level, which follows the transformation of mesopic luminous efficacy function from photopic to scotopic vision. In addition, the peak wavelength located at 450 nm is beneficial to the larger color gamut volume.

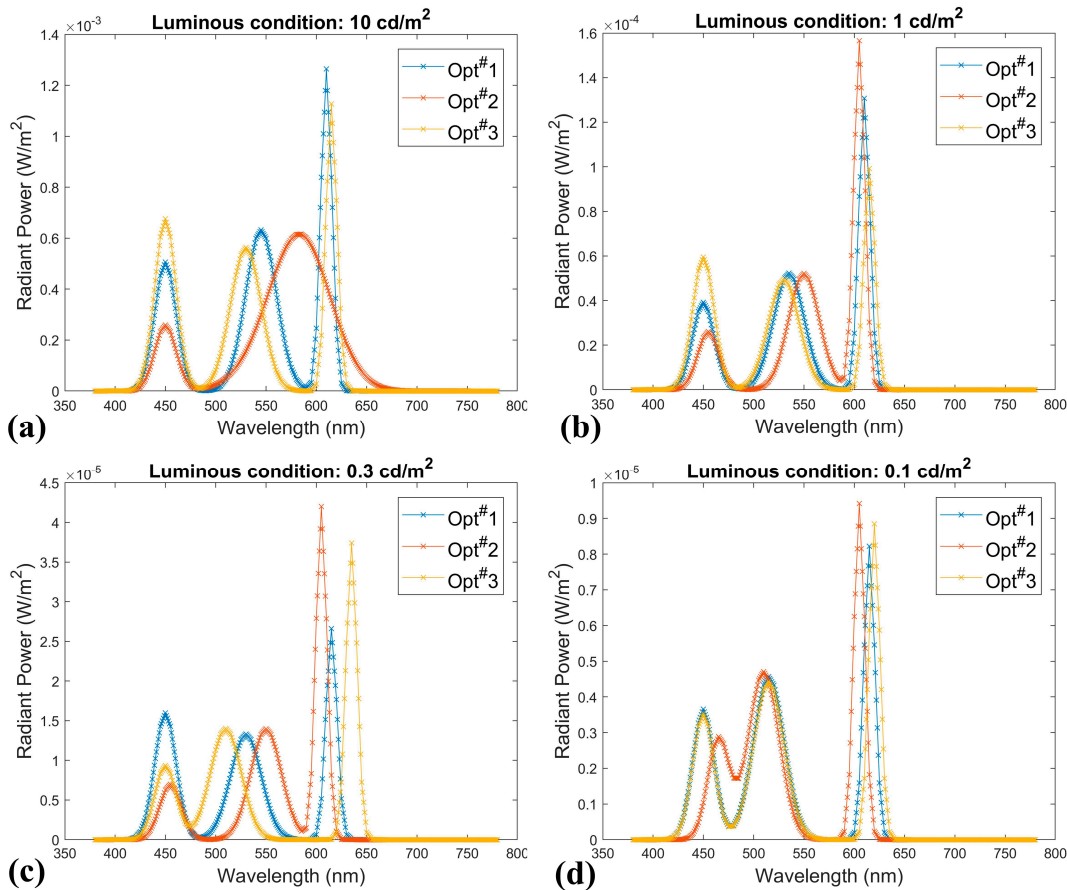

**Figure 2.** Optimal spectral power distributions (SPDs) for four lighting conditions: (**a**) 10 cd/m$^2$, (**b**) 1 cd/m$^2$, (**c**) 0.3 cd/m$^2$, (**d**) 0.1 cd/m$^2$.

Table 3 shows the corresponding values of mesopic luminance, gamut volume, and S/P ratio. A trade-off between mesopic luminance and color gamut is quite significant. The photopic/mesopic luminous efficacy of radiation (LER) of equal energy white light is about 181, 206, 230, 259 lm/W for 10, 1, 0.3, and 0.1, cd/m$^2$, respectively. In the aspect of the S/P ratio, the optimal spectra with a higher

S/P ratio can raise the mesopic luminance and also extend the color gamut volume with a decrease in luminance. In other words, they are more appropriate for application in mesopic vision. The outcome is in accordance with the previous study [14]. Figure 3 shows the color gamut of optimal spectra in the CIE LAB color space under each lighting condition.

**Table 3.** Information of optimal spectra.

| cd/m² | Luminance (cd/m²) | | | Color Gamut Volume (×10³) | | | S/P Ratio | | |
|---|---|---|---|---|---|---|---|---|---|
| | Opt#1 | Opt#2 | Opt#3 | Opt#1 | Opt#2 | Opt#3 | Opt#1 | Opt#2 | Opt#3 |
| 10 | 21.79 | 24.52 | 17.16 | 97.40 | 51.40 | 122.98 | 1.57 | 0.94 | 2.46 |
| 1 | 1.92 | 2.17 | 1.67 | 20.60 | 12.71 | 26.61 | 1.78 | 1.10 | 2.46 |
| 0.3 | 0.50 | 0.59 | 0.42 | 11.84 | 4.59 | 13.17 | 2.46 | 1.10 | 3.72 |
| 0.1 | 0.18 | 0.19 | 0.17 | 4.16 | 2.70 | 4.30 | 3.16 | 3.09 | 3.20 |

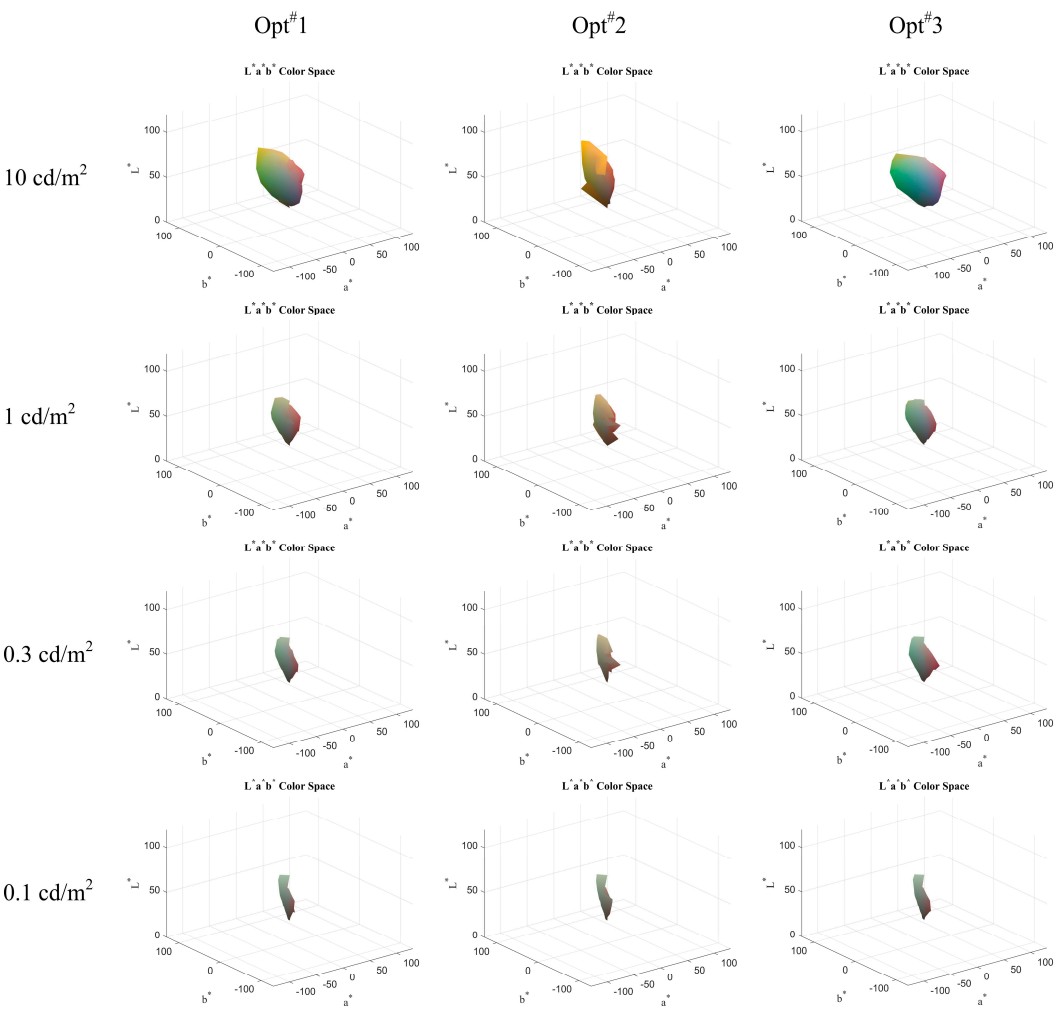

**Figure 3.** Illustration of color gamut of optimal SPDs.

The information of optimal spectra inclusive of LER, CRI, $R_f$, $R_g$, CCT, and Duv is presented in Table 4. LER($m$) can be described as Equation (14), where LER$_{mes}$($m$) is calculated only when the luminance $L_p$ is below 5 cd/m². Otherwise, LER$_p$ is used to represent the LER. $R_f$ and $R_g$ are the fidelity and gamut index defined from IES TM-30-15 to evaluate color rendition. At present, the majority of the white LED spectrum is optimized based on luminous efficiency and color rendering index mostly in the photopic vision, which shows a wider color gamut than in mesopic vision. However, the conventional color rendering properties such as CRI and IES TM-30-15 cannot express the color variation with the

decreasing of illuminance in dim lighting conditions. Referring to the color vector graphic, as shown in Figure 4, the optimal spectra Opt#1 has better performance in color gamut as the $R_g$ values increase. Additionally, due to the positive correlation between the S/P ratio and CCT [13], the optimal spectra with higher CCT consequently can improve the color perception, especially under low-light levels.

$$\text{LER}_{\text{mes}}(m) = \frac{m + (1-m)(\text{S/P})(683/1699)}{m + (1-m)(683/1699)} \times \text{LER}_{\text{p}}, \ \text{LER}_{\text{p}} = \frac{\int_{380}^{780} 683 V(\lambda) S(\lambda) d\lambda}{\int_{380}^{780} S(\lambda) d\lambda} \tag{14}$$

**Table 4.** Information of optimal spectra.

|  | **10 cd/m$^2$** | **1 cd/m$^2$** | **0.3 cd/m$^2$** | **0.1 cd/m$^2$** |
|---|---|---|---|---|
| LER$_{\text{Opt}^\#1}$ | 395 | 394 | 387 | 450 |
| LER$_{\text{Opt}^\#2}$ | 444 | 446 | 450 | 480 |
| LER$_{\text{Opt}^\#3}$ | 311 | 343 | 319 | 436 |
| CRI$_{\text{Opt}^\#1}$ | 74 | 79 | 63 | 42 |
| CRI$_{\text{Opt}^\#2}$ | 41 | 62 | 62 | 39 |
| CRI$_{\text{Opt}^\#3}$ | 63 | 63 | -8 | 27 |
| $R_{f,\text{Opt}^\#1}$ | 66 | 76 | 71 | 54 |
| $R_{f,\text{Opt}^\#2}$ | 44 | 60 | 60 | 44 |
| $R_{f,\text{Opt}^\#3}$ | 71 | 71 | 24 | 47 |
| $R_{g,\text{Opt}^\#1}$ | 99 | 111 | 119 | 117 |
| $R_{g,\text{Opt}^\#2}$ | 76 | 90 | 80 | 86 |
| $R_{g,\text{Opt}^\#3}$ | 119 | 119 | 121 | 122 |
| CCT$_{\text{Opt}^\#1}$ | 4607 | 4350 | 7456 | 7394 |
| CCT$_{\text{Opt}^\#2}$ | 3036 | 3083 | 3083 | 5314 |
| CCT$_{\text{Opt}^\#3}$ | 7456 | 7456 | 7975 | 7248 |
| Duv$_{\text{Opt}^\#1}$ | 0.0046 | 0.0029 | 0.0010 | 0.0042 |
| Duv$_{\text{Opt}^\#2}$ | 0.0048 | 0.0032 | 0.0032 | 0.0045 |
| Duv$_{\text{Opt}^\#3}$ | 0.0010 | 0.0010 | 0.0006 | 0.0006 |

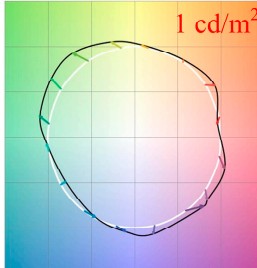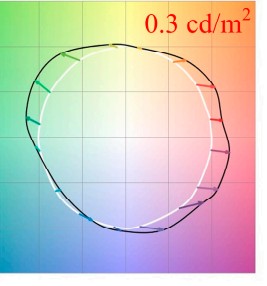

**Figure 4.** Color vector graphic of Opt#1 optimal SPDs.

### 3.2. Characteristic of White LED Spectra

For better understanding of the characteristic of pc-WLED on mesopic luminance and gamut volume, the optimal spectra Opt#1 with the variance of peak wavelength shifts and the different power density ratios (i.e., the rate of the maximum powers of blue, green, and red components) of three components were investigated under 10 and 0.1 cd/m$^2$ luminance levels, respectively. Note that the luminance level is referred to as the luminance of an equal energy white. The test SPDs were normalized to have the equivalent total radiance to the equal energy white. As shown in Figure 5; Figure 6, the results of test spectra where the peak wavelength is shifted from 450 to 480 nm for

blue-chip, 500 to 550 nm for green phosphor, and 605 to 645 for red phosphor with 5 nm interval, and power density ratios ranging from 0.1 to 1.0, are depicted.

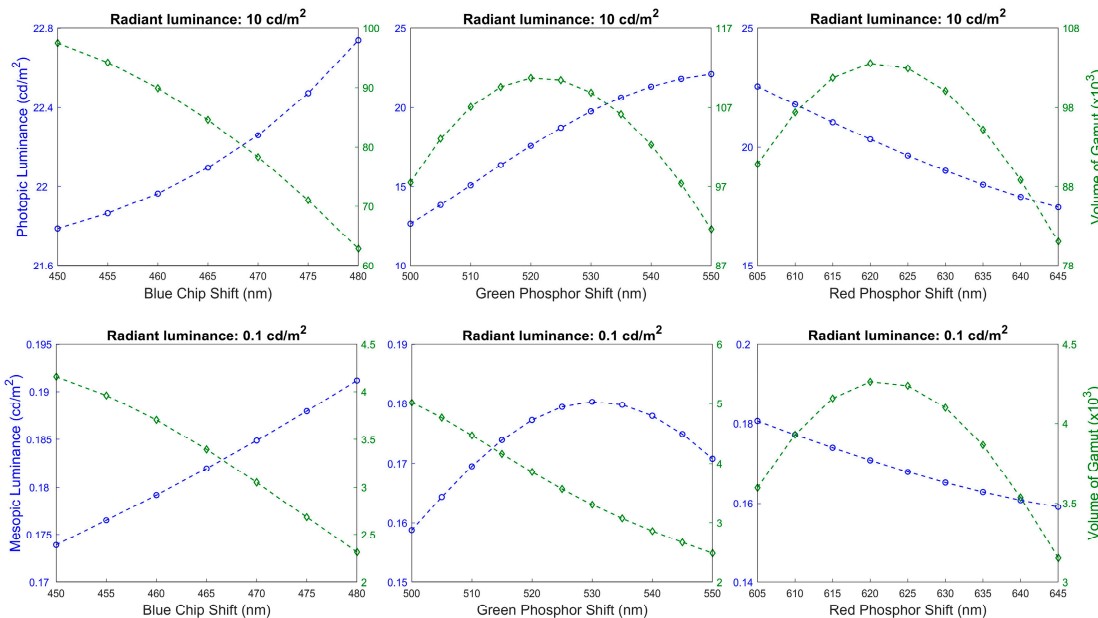

**Figure 5.** Characteristic of luminance and gamut volume of peak wavelength shifts (left to right: blue, green, and red).

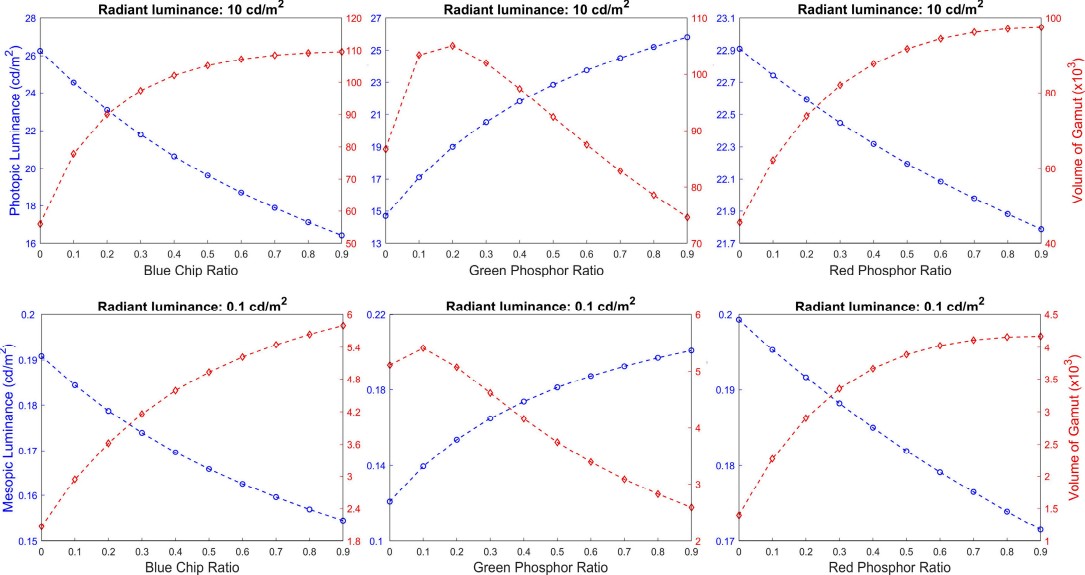

**Figure 6.** Characteristic of luminance and gamut volume of different power density ratios (left to right: blue, green, and red).

According to the simulation results, it indicates that the mesopic luminance will increase with the shifts of the blue peak from a shorter wavelength to the longer ones, but the color gamut volume shows the opposite trend under these two luminance levels. Besides, the two lighting levels present the same tendency except for the wavelength shift of the green peak, which is due to the conversion of luminous efficacy. For the density ratio, the mesopic luminance increases with a higher rate of green. Moreover, the higher power of blue and red components contributes to the extension of the color gamut volume.

### 3.3. Mesopic Color Reproduction

The mesopic color reproduction can be carried out by following the procedure, as shown in Figure 7. A test image is captured using a Canon EOS 7D DSLR camera under a light booth with a D65 light source under 318 cd/m$^2$ ($\cong$1000 lux). The image in CR2 format is converted to a 16-bit tiff format in linear RGB (gamma = 1) to minimize the impact of tone enhancement while ensuring wide color gamut. A third-order polynomial regression as Equation (15) is used to correlate the linear RGB value to *X, Y, Z*, and *V′*($\lambda$) values as the colorimetric model. The simulated image of optimal SPDs under four luminance levels is shown in Figure 8.

$$\begin{bmatrix} X \\ Y \\ Z \\ V' \end{bmatrix} = M_{4\times14}\begin{bmatrix} R^3 & G^3 & B^3 & R^2 & G^2 & B^2 & R & G & B & RG & GB & RB & RGB & 1 \end{bmatrix}^T \quad (15)$$

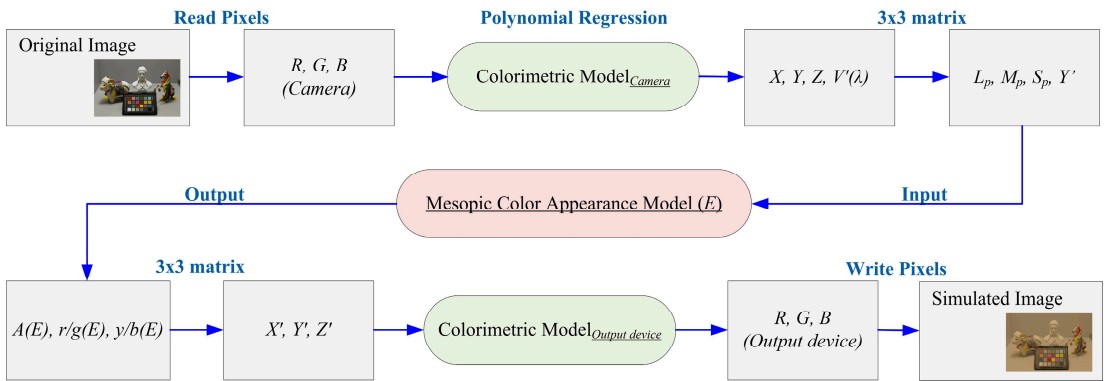

**Figure 7.** Flow chart of the mesopic color reproduction.

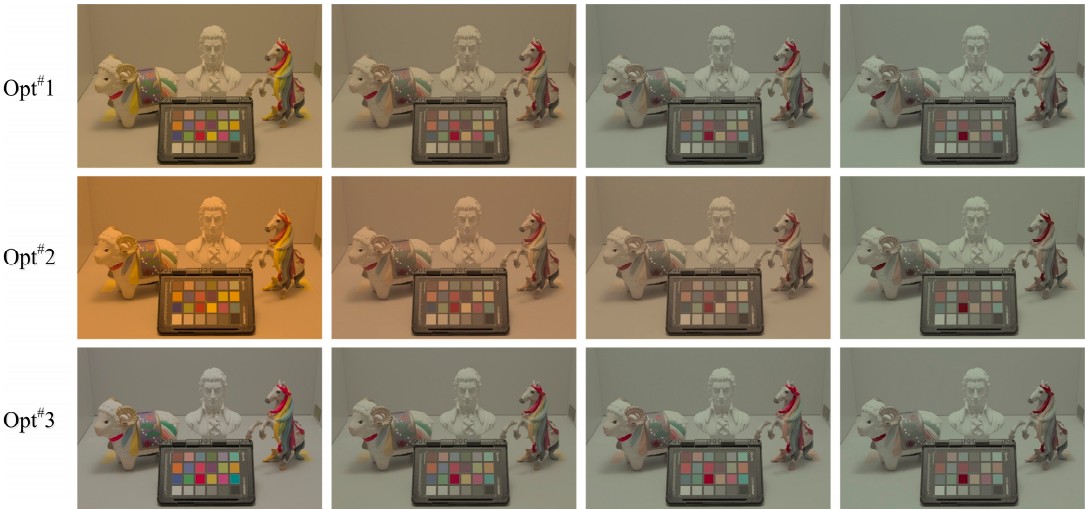

**Figure 8.** Simulated images of optimal SPDs (left to right: 10, 1, 0.3, and 0.1 cd/m$^2$).

## 4. Evaluation Model

For calculating not only the mesopic luminance but the color gamut volume, a large number of computational resources is required, such as the mesopic luminance using iteration method and color gamut volume by a mesopic color appearance model and the Delaunay triangulation. Thus, an evaluation model is essential to offer a faster assessment of lighting quality under dim lighting conditions for lighting applications.

### 4.1. Modeling

To train a neural network, the SPDs of white LED spectra, as described in Section 2.1 under 10, 1, 0.3, and 0.1 cd/m$^2$ are regarded as the inputs, and the outputs include the S/P ratio, mesopic luminance, and color gamut volume. In total, 358,924 (89,731 records × 4 luminance levels) SPDs are collected. In addition, the min-max scaling is used to normalize the input data for improving accuracy and convergence.

The architecture of the evaluation model is a 1D convolutional neural network (1D CNN), as shown in Figure 9, containing a convolutional layer, fully connected layer, two hidden layers with 50 neurons, and an output layer with three output values and a linear activation function to solve the regression task. The convolutional layer with an input shape of 81 × 1 and one stride was used to extract feature maps by 32 filters (10 × 1) without padding, and a hyperbolic tangent (Tanh) activation function was added to each layer. The optimization algorithm was achieved by adaptive moment estimation (Adam), which combines the advantages of Momentum and the RMSProp algorithm can adaptively accelerate gradient vectors and adjust the learning rate based on the root mean square of weights for training speed up. Additionally, the number of epochs and batch size were set as 500 and 256 when training.

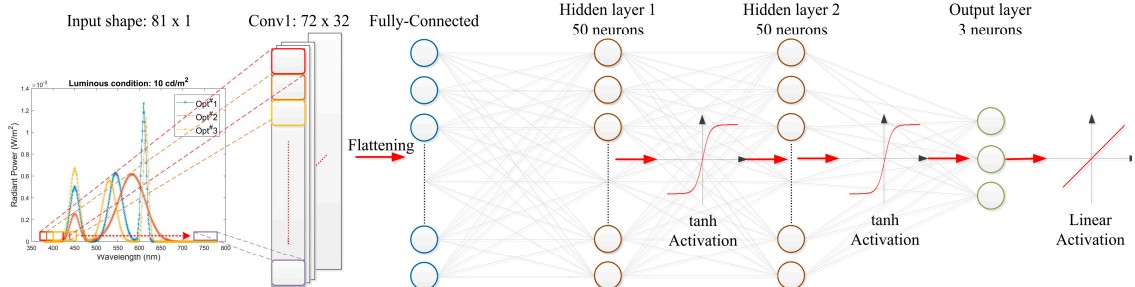

**Figure 9.** Architecture of the evaluation model.

### 4.2. Model Evaluation

In the field of machine learning, *k*-fold cross-validation (CV), especially the 10-fold CV, is most commonly used to verify the performance of a designed model. The parameter called *k* randomly splits the data into *k* subsets and one of the subsets is chosen for testing with the remaining $k - 1$ subsets for training. The procedure is repeated until each subset is treated as a testing set.

For regression analysis, the performance of a model is evaluated using the coefficient of determination (R$^2$), root-mean-square error (RMSE), and mean absolute error (MAE). The average and standard deviation of the testing results are listed in Table 5, which indicates the performance and stability of the model, respectively. Figure 10 shows the scatter plots of the predicted values and the ground-truth where the R$^2$ scores are 0.99. As a consequence, the 1D CNN model predicts mesopic luminance, color gamut volume, and S/P ratio with high R-squared, low RMSE, and MAE values.

**Table 5.** Evaluation metrics for machine learning model.

| ($\times 10^{-2}$) | Mesopic Luminance | Color Gamut Volume | S/P Ratio |
|---|---|---|---|
| Mean RMSE ± SD | 1.6025 ± 0.9088 | 5.2161 ± 4.3522 | 1.2786 ± 0.1880 |
| Mean R$^2$ ± SD | 99.999 ± 0.0010 | 99.999 ± 0.0006 | 99.955 ± 0.0128 |
| Mean MAE ± SD | 0.9343 ± 0.4938 | 3.1094 ± 2.0530 | 0.8929 ± 0.1710 |

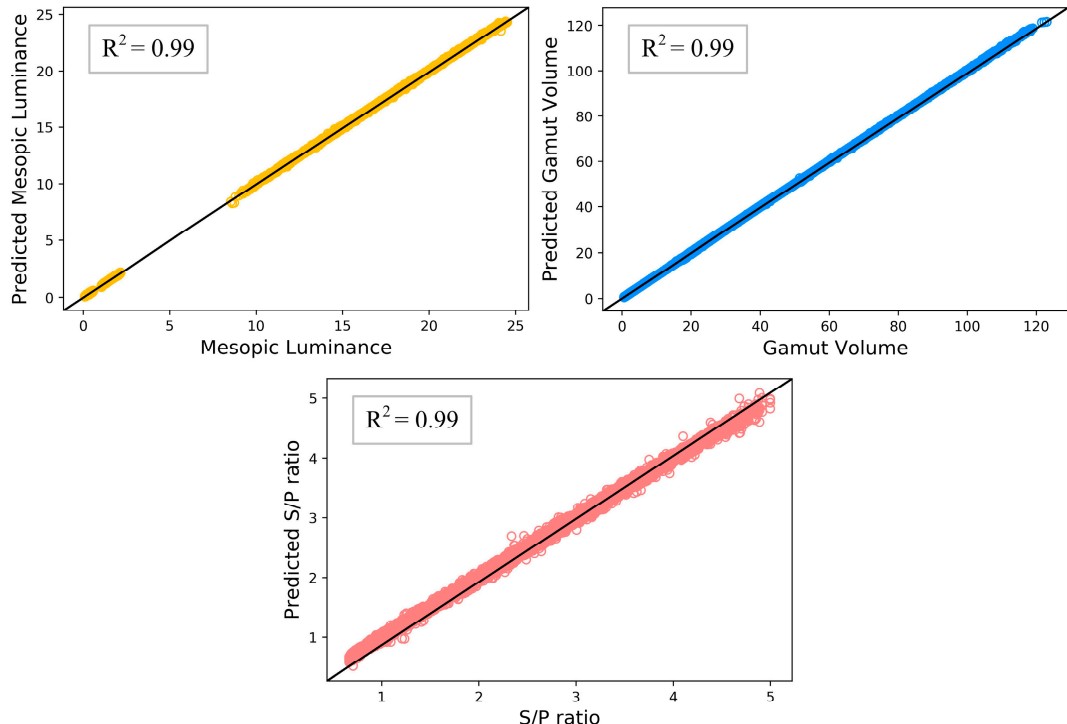

**Figure 10.** Scatter plots of the predictions.

## 5. Conclusions

Three models were examined with different peak wavelength positions and density ratios to achieve optimal white LED spectra based on mesopic luminance and color gamut volume. The modified MOVE-model and mesopic color appearance model were adopted to calculate the mesopic luminance and present perceptual color attributes. For energy savings, reductions, and color perception enhancement, optimal spectra that reach relatively higher mesopic luminance and color gamut volume, highest mesopic luminance, and largest gamut volume were obtained with a Duv of ±0.0054 and the CCT ranging from 2000 to 8000 K. With the decrease of luminance level, the spectra with higher CCT and S/P-ratio increase the mesopic luminance and also extend the range of color gamut volume. For this characteristic, the higher density ratio of blue and red components results in a better performance of color gamut volume. However, there is a trade-off between mesopic luminance and color gamut volume. Furthermore, an evaluation model derived from the 1D convolutional neural network can be used to predict mesopic luminance, color gamut volume, and S/P ratio accurately for lighting application with high R-squared, and low RMSE and MAE values.

**Author Contributions:** H.-C.L., was responsible for simulation, optimizing the design, organization, and writing—manuscript preparation. P.-L.S., provided the concept and undertook project administration. P.-L.S., Y.H., and M.R.L. were responsible for supervision and writing—review and editing. All authors have read and agreed to the published version of the manuscript.

**Funding:** This research received no external funding

**Conflicts of Interest:** The authors declare no conflict of interest.

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
