# Peer review of "Spectral Optimization of White LED Based on Mesopic Luminance and Color Gamut Volume for Dim Lighting Conditions"

_applsci, doi:10.3390/app10103579_

Round 1

Reviewer 1 Report

General considerations

The goal of the paper is to propose optimal white LED spectra and predict the mesopic luminance through the use of a neural network.

Despite a wide introduction, lacks references of other works on neural network used to predict the Spectral Power Distribution. In fact, there are not comparison with the performance of models proposed in literature.

The strengths of the paper respect to the other works in the literature are not highlighted. What is the innovation of the results compared to the other works present in the literature? Moreover the introduction is badly written and insufficient exposure clarity.

There are differents typing errors in the papers. A list is provided below of some of those errors.

In the paper is common use the acronym before it is defined (such as CQS, CRI, SPD, ... ), instead others are not defined (such as CIE, IES, S/P, ... ).

Row 32-33: It is necessary to review this sentence. At m=0 the curve corresponds to the scotopic spectral luminance efficacy, and at m=1 the curve corresponds to the photopic spectral luminance efficacy? Explain the relation between Lmes and the values 5 and 0.005 cd/m^2 as a function of m. m is 0 if Lmes is greater 5cd/m^2 , and m is 1 if Lmes is smaller than 0.005 cd/m^2.

Row 51: Define the three visual tasks.

Row 52: Replace cd m^2 with cd/m^2.

Row 53: Are achromatic threshold and reaction time the two items? Specify them.

Row 59: What is the upper luminance limit of the modified MOVE-model?

Row 66: What is the B coefficient? Describe in reference to the equation Emes=B·Ep, where Ep is photopic illumination and Emes is mesopic equivalent illumination.

Row 101: Replace matric with metric.

Row 103: CCT already defined in row 67.

Row 258: In equation (13) the x corresponds to m_(n-1) in equation (8), explain.

Row 279: In Fig. 6 the mesopic luminance decreases with a higher rate of red phosphor.

Figure 5: The y axis is missing for mesopic luminance in function of green phosphor shift.

Figure 6: Is photopic luminance the y label in 10 cd/m^2 radiant luminance?

Row 292: value and to X, Y, Z, and …

Row 347: What do you mean by 'acceptable'? In what terms acceptable?

Finally it is interesting to add comparisons between the original and the predict SPD, and the estimation error of the colorimetric quantities of light sources.

Reviewer 2 Report

Dear authors,

I congratulate you for the methodology and significance of the presented paper. Working myself in the domain, I would nevertheless ask you to perform a few adjustments to make your paper easier to read.

  • the abstract should describe a wider context. There are two main aspects: if you specify street lighting systems in terms of mesopic values, it will be possible to reduce the amount of produced light, meaning energy reduction and diminution of light pollution. The abstract should also specify what is the problem you are trying to address : why do you need a new model ?
  • the introduction provides an extensive coverage about mesopic luminance. The abstract indicates two parameters are optimized: luminance and gamut volume. It would be interesting to provide some info about the visual advantages of a wider gamut.
  • 2.1: Eq 3 is not in its usual form; it should read g(x) = A exp(-0.5 ((x-x0)/a)^n). From inspection, you have normalized all spectra, meaning A = 1; but x0 is the mean (or central wavelength) and should be kept. Furthermore, I don't like the 'abs' function which may introduce discontinuities.
  • 2.1, line 153-155: this is really difficult to understand. What are 'components' ? How do you define "Power Density Ratio" ? How are MS1, MS2, MS3 defined ? What about records: is that the total number of components, or combination of component * current/luminance level * spectral points + ... ?
  • 2.2, eq.6: you try to find the fixed point of an iterative process. Why, how, what is the equilibrium equation behind it ?
  • 2.2, eq. 9 to 11: I'm lost. Could you provide some theoretical background ? What is the purpose ? Furthermore, in fig.1, the box with A(E) and others has no source.
  • 2.3, eq. 12: the criterion is the product of two contrast functions. Please say a few words about the choice of such structure. What is the search space leading to Lmax and Lmin ? Is it about one device or the whole set of devices and conditions ? In such case, how many different points do you have ?
  • 3.1, figure 2: graphs are a bit strange. It looks like the line width is not the same between 600 and 650 nm ? About table 3, either I do not well understand the index, either the units for the first group of data are missing. Are you comparing photopic with mesopic luminances ? If yes, make it clearer. If no, explain. Table 4: please define Rf and Rg
  • 3.2: Fig.5, what is the difference between the index, "radiant luminance", and the photopic luminance ? Why is the photopic luminance always greater than the radiant luminance ?  To me, there is an issue as all graphs in Fig.6 have "mesopic luminance as ylabel. In Fig.5: there should be one ylabel on the extreme left (luminance) and one secondary ylabel (gamut volume) on the extreme right, no need to repeat 3 times. The graph at the middle of the second row lacks yticks on the left side. For fig. 6, in the middle column, there is a break around 0.1 .. 0.2 You should compute a few more points in the vicinity to get a smoother curve
  • 4: say a few words about why you need a model. Is the relationship too non-linear ? Does one evaluation require too much computing power ? ...
  • For 4.2, table 5. GUM (Guide of uncertainty measurements) suggest to express SD with 2 significant decimal figures if the first one is "1" and with 1 otherwise. Then you should give some confidence interval by multiplying SD by 2. The third column, 1st value; mean 1.2786 and SD 0.094 should be given as (1.2786 +/- .188) => 1.28 +- 0.19 (1.09, 1.47)
  • The conclusion should also encompass a wider perspective, like f.i. the potential for energy savings. A simple scenario is to assume a street lighting pole with 70W injected into the LEDs

Regards
